# Deep-Asymmetry: Asymmetry Matrix Image for Deep Learning Method in Pre-Screening Depression

**DOI:** 10.3390/s20226526

**Published:** 2020-11-15

**Authors:** Min Kang, Hyunjin Kwon, Jin-Hyeok Park, Seokhwan Kang, Youngho Lee

**Affiliations:** 1Department of Computer Engineering, Gachon University, Sungnam-si 13306, Korea; km8846@gachon.ac.kr (M.K.); shkang@gachon.ac.kr (S.K.); 2Department of IT Convergence Engineering, Gachon University, Sungnam-si 13306, Korea; euleekwon@gmail.com (H.K.); rev.hyeok@gmail.com (J.-H.P.)

**Keywords:** electroencephalogram, major depressive disorder, convolutional neural networks, deep learning, asymmetry, asymmetry image

## Abstract

To have an objective depression diagnosis, numerous studies based on machine learning and deep learning using electroencephalogram (EEG) have been conducted. Most studies depend on one-dimensional raw data and required fine feature extraction. To solve this problem, in the EEG visualization research field, short-time Fourier transform (STFT), wavelet, and coherence commonly used as method s for transferring EEG data to 2D images. However, we devised a new way from the concept that EEG’s asymmetry was considered one of the major biomarkers of depression. This study proposes a deep-asymmetry methodology that converts the EEG’s asymmetry feature into a matrix image and uses it as input to a convolutional neural network. The asymmetry matrix image in the alpha band achieved 98.85% accuracy and outperformed most of the methods presented in previous studies. This study indicates that the proposed method can be an effective tool for pre-screening major depressive disorder patients.

## 1. Introduction

Major depressive disorder (MDD), commonly called depression, is a neurological disorder that affects more than 3 million people worldwide every year [1]. Depression negatively affects the way a person feels, thinks, and behaves. It seriously disrupts the person’s social life, and in severe cases, leads to serious problems, such as suicide [2]. There are many ways to diagnose depression; however, the most common approach is through consultation with a psychologist or psychiatrist [3]. The diagnostic and statistical manual of mental disorders (DSM-V) [4], Beck depression inventory [5], Hamilton depression rating scale tests [6], and other supplementary indicators can be used.

However, diagnosis of depression through subjective judgment of a person is limited. For example, if the symptoms of depression are masked as symptoms of disorders other than depression, they may not be judged as depression, and thus, may be misdiagnosed. In particular, referred to as masked depression [7,8] appear as symptoms other than depression. Such atypical depression is difficult to diagnose and easy to misdiagnose, but it carries a great risk [9]. In these cases, there is a problem that the disease may be getting worse because they did not receive treatment at the right time. In addition, many depressed patients are concerned about the public perception of their mental illness; hence, they do not easily reveal their condition to experts [10].

Therefore, there is a need for a consistent and accurate method for classifying depressed patients and healthy individuals. Several studies have attempted to identify biomarkers of depression based on objective physiological data such as computed tomography (CT), functional magnetic resonance imaging (FMRI), and electroencephalogram (EEG) [11]. In particular, the noninvasive method of acquiring EEG by attaching electrodes to a patient’s head has gained attention as it is more cost-effective and easier to implement than conventional tracking methods, such as CT and FMRI.

Research on EEG-based depression diagnosis using machine learning technology is of great interest. Various features extracted from EEG are entered into a machine learning model, and the model is trained to classify MDD patients and healthy controls effectively. Mumtaz et al. proposed a machine learning model that classifies MDD patients based on features such as frequency band power and alpha hemisphere asymmetry [12]. Mahato et al. presented a machine learning model that uses band power, theta asymmetry, and alpha1 and alpha2 features [13]. In addition, Mahato et al. also presented a classification model that uses linear features, such as band power and hemispheric asymmetry, and nonlinear features, such as relative wavelet energy (RWE) and wavelet entropy (WE) [14].

One of the features commonly shown in research results is brain asymmetry. Both Allen et al. [15] and Dharmadhikari et al. [16] reported that frontal lobe asymmetry is a promising biomarker of depression.

In previous studies, deep learning techniques were also used to classify brain waves of depressed patients. Deep learning is widely used in medical diagnosis owing to its wide range of functions—from pre-processing to extraction of key features based on self-learned data. Acharya et al. presented a 13-layer model for EEG-based depression screening using a convolutional neural network (CNN) [17]. Mumtaz et al. proposed one-dimensional CNN (1D CNN) and 1DCNN + LSTM (long short-term memory) models to confirm the possibility of using deep learning architectures for diagnosis of depression [18]. In addition, Betul Ay et al. confirmed the effective classification performance of raw EEG signals obtained from the left and right hemispheres of the brain using CNN + LSTM [19].

This study proposes a deep-asymmetry method that converts brain asymmetry, which has been identified as a promising biomarker, into a matrix-type image and provides it to a convolution neural network. Figure 1 illustrates the proposed method. The asymmetry image matrix can express the difference in the degree of asymmetry between different channels. The model automatically extracts image features through a CNN and classifies MDD patients and healthy controls using a fully-connected network. The proposed approach was tested on an open-accessible dataset and its performance was compared with previous related studies.

## 2. Materials and Methods

### 2.1. Dataset

This study used the EEG data set [20], an open access data set, collected by Mumtaz et al. at the Hospital Universiti Sains Malaysia (HUSM). The data set comprises information of 34 depressed patients (mean age = 40.33, SD = ±12.861) and 30 healthy controls (mean age = 38.227, SD = ±15.64). The MDD patients were diagnosed with DSM-IV and underwent a two-week drug washing period before EEG collection. The experimental setup and data collection of the study were approved by the ethics committee of HUSM and written consents were obtained from the subjects. The EEG data collection was based on the international 10–20 system [21]. A total of 19 channels were collected, including frontal (Fp1, F3, F7, Fz, Fp2, F4, and F8), central (C3, C4, and Cz), parietal lobes (P3, Pz, and P4), occipital (O1 and O2), and temporal regions (T3, T4, T5, and T6). The data were filtered between 0.5–70.0 Hz and a 50 Hz notch filter was used to suppress power line noise. The EEG data were collected in three states: eyes closed (5 min), eyes open (5 min), and with a given specific visual stimulus (10 min). The EEG data were collected with a sample rate of 256 samples per second. This study used the eye-closed (EC) data and eye-opened (EO) data in the HUSM data set.

### 2.2. Data Preprocessing

The EEG data were normalized before use so that each channel would have similar amplitude scaling. Each channel was normalized using the min-max normalization method. Electroencephalogram data are easily affected by noise, such as eye blinking and muscle movement. Therefore, independent component analysis (ICA) was used to further remove EEG noise. ICA is an effective method for removing artifacts [22] and has been adopted in previous studies [10].

The number of samples in a dataset is an important consideration in machine learning problems. This is a commonly considered problem in other EEG studies [23]. Data segmentation, the process of dividing samples in EEG data into meaningful segments, is a potential approach to address this problem, and it has been used in existing EEG studies [24]. Therefore, in this study, the 5 min data set was divided into epochs of 4 s (1024 samples) each and assigned with the same label.

### 2.3. Visualizing EEG Image of Brain Asymmetry

The asymmetry score can be obtained through the difference between observations, such as the EEG signal power of each channel [25]. The relative powers of delta (0.5–4 Hz), theta (4–8 Hz), alpha (8–13 Hz), and beta (13–30 Hz) were calculated to obtain asymmetry images for each EEG band. The power spectrum (PSD) of the EEG signal was calculated using Welch’s periodogram [26] The window length was set to two times the inverse of the lower frequency of interest and overlap was set to 50% This was used to obtain power spectral density S. The calculated frequency band power spectral density was a parabolic approximation using the Simpson’s method [27]. Equations (1) and (2) were used to calculate the relative power of each channel at the desired frequency.
(1)Rpch1= ∑f=f1f2Sch1∑f=0.5Hz30HzSch1
(2)Rpch2= ∑f=f1f2Sch2∑f=0.5Hz30HzSch2
where, f1 and f2 denote the lowest and highest frequencies in the band, respectively. For example, if the relative EEG signal power of the alpha wave (8–13 Hz) is calculated, f1 = 8 and f2 = 13. The power spectral densities at each channel are denoted as Sch1 and Sch2 Equation (3) is used to obtain the power asymmetry between channels ch1 and ch2. The parameter A(ch1, ch2) is calculated as the difference between the relative power values of ch1 and ch2. For example, if fp1 is ch1, then ch2 is set to one of the 16 channels (Fp1, F3, C3, P3, O1, F7, T3, T5, Fp2, F4, C4, P4, O2, F8, T4, and T6).
(3)$$A\left(\mathrm{ch}1,\mathrm{ch}2\right)=\frac{{\mathrm{Rp}}_{\mathrm{ch}1}\;-\;{\mathrm{Rp}}_{\mathrm{ch}2}}{{\mathrm{Rp}}_{\mathrm{ch}1}\;+\;{\mathrm{Rp}}_{\mathrm{ch}2}}$$

Using Equation (3), it is possible to calculate the difference between the relative power of two channels for each channel pair, and hence, the difference in activity corresponding to each part of the brain can be identified. Figure 2 shows an example of EEG asymmetry calculation.

A (ch1, ch2) can have a value between −1 and 1. A positive value means that ch1 has a relatively higher EEG relative power. A negative value means that ch2 has a relatively lower EEG relative power. Based on this, the asymmetry matrix image shown in Figure 3 is proposed to express the value of A (ch1, ch2) as one image.

The image is composed of 16 rows and columns, and each row and column correspond to 16 channels. Each cell has a color corresponding to A (ch1, ch2) value, as indicated on the color bar on the right. Red indicates an A (ch1, ch2) value close to +1, while blue indicates a value close to −1. Clear reds and blues indicate a greater difference between the channels and a more active brain. Yellow and green indicate that the relative power between the channels is similar.

### 2.4. Classification Model

This study proposes a deep learning model using a CNN to classify depressed patients and normal controls using asymmetry matrix images. Table 1 lists the parameters of the model. A 3-layer model comprising a 2-dimensional convolution layer and a pooling layer followed by a fully connected layer was used. The convolution layer used the ReLU activation function. A sigmoid function was used in the output layer. In addition, batch normalization [28] was performed to prevent gradient vanishing problem during model training and a dropout layer [29] was used to prevent overfitting of the model. The dimensions of the input image were set to 64 × 64 × 3. Image input was carried out using a mini batch, which was used to improve the model training performance and speed [30]. When training the model, the learning rate was set to 0.0001 and 10 epochs were used. The binary cross-entropy loss function was used and training was performed using Adam optimizer [31].

### 2.5. Evaluation

For model evaluation, a k-fold cross-validation [32] (k = 5) was used. When dividing the data set into training and testing sets, the data set was randomly mixed and then divided into 5 subsets to avoid data bias. One subset was used for testing and the other four subsets were used for training. The model was evaluated using the average of the evaluation indicators obtained after 5 repetitions without overlapping.

In this study, accuracy (Equation (4)), sensitivity (Equation (5)), and specificity (Equation (6)) based on the confusion matrix were used as indicators of performance evaluation. Sensitivity was defined as the percentage of actual MDD patients in all cases with MDD patients (*TP* + *FN*), and specificity was defined as the percentage of actual healthy controls in all cases with healthy controls (*TN* + *FP*). Accuracy was defined as the percentage of correctly classified MDD patients and healthy controls.
(4)$$Accuracy\;=\;\frac{TP\;+\;TN}{TP\;+\;TN\;+\;FP\;+\;TN}$$
(5)$$\mathrm{sensitivity}=\;\frac{TP}{TP\;+\;FN}$$
(6)$$\mathrm{specificity}=\;\frac{TN}{TN\;+\;FP}$$

Additionally, evaluation was conducted using the receiver operating characteristic (ROC) curve, which is widely used in the evaluation of binary classification. The ROC curve evaluates sensitivity and specificity based on several thresholds.

## 3. Results

### 3.1. Visualized EEG Image of Brain Asymmetry

In this study, a matrix image of the degree of asymmetry for each channel was created based on the relative signal power in the delta (0.5–4 Hz), theta (4–8 Hz), alpha (8–13 Hz), and beta (13–30 Hz) bands. Each model with this input image was constructed and its performance was compared. Figure 4, Figure 5, Figure 6 and Figure 7 shows example images of healthy controls and MDD patients based on the frequency of each band.

### 3.2. Classification Model

Table 2 shows the performance of the asymmetry matrix image-based classification model in EC dataset. The table presents the average of each evaluation index from the five repetitions. The table shows that the best performance was obtained using the alpha band-based asymmetry image, which had an accuracy of 98.85, sensitivity of 98.84, and specificity of 98.66.

Table 3 shows the performance of the asymmetry matrix image-based classification model in EO dataset. From the table, the accuracy of the model using alpha band-based asymmetry image was 97.56, sensitivity was 98.75, and specificity was 96.31.

Figure 8 and Figure 9 shows the performance of the proposed model under each band using the ROC curve. The figure shows the graph of each five cross-validations as well as the graph of the average value. The alpha band had the best ROC curve compared with the other bands. The area under its curve (AUC) also showed effective classification performance with an average of 0.9879 in EC dataset’s alpha band.

## 4. Discussion

It is important to diagnose and treat depression early. However, many people are reluctant to seek medical help for their illness as they are concerned about the public perception of mental illness. As EEG devices become popular and the public can easily use them, the EEG-based machine learning and deep learning methods can be used as powerful tools for identifying depression and providing follow-up treatment.

One of the major novelties of this study is converting the brain asymmetry matrix into a 2D image and providing it to a 2D CNN model. Several methods for the conversion of 1D signal data into a 2D image have been considered. For example, the STFT, wavelet-based spectrogram method [33], and brain coherence network-based method [34,35] have been proposed. However, to implement an effective 2D CNN image, this study used brain asymmetry, which has been identified as an important biomarker of depression. The matrix was expressed as a visualized image. A high classification accuracy of 98.85% was achieved using the alpha asymmetry image. This is consistent with the findings of other studies published in the literature that the alpha asymmetry feature can be used as a biomarker for reading depression [11].

Table 4 summarizes the methodology and results of the studies using the same dataset. Accuracy, a commonly used index in other studies, was used. Table 4 shows that the proposed method surpasses all the other methods presented in the table, except for the study conducted by Saeedi et al. Thus, compared with the other methods using raw EEG signals, the asymmetry matrix image method proved to be effective for detecting and controlling depression.

Similar to other studies, the size of the data set is the limitation of the study. Because small datasets are at risk of overfitting. It is also because a smaller sample size limits generalization to a larger population. As our small pilot study, generalization of the proposed classification model requires further evaluation with more study samples.

So we willingly released the code. Our code can be accessed from Appendix A. Our efforts will show that the model can be validated on more test data and the methodology can be further advanced.

Furthermore, beyond the classification of depressed patients and healthy controls, it would be meaningful to find the correlation of the proposed classification function with scores such as beck’s depression inventory (BDI) score.

However, to overcome model overfitting problem, data segmentation, batch normalization, and dropout layer were used in this study. Future research will be focused on generalizing results and presenting a more reliable model using more samples and other types of EEG data.

## 5. Conclusions

Prescreening and treatment of depression are very important. Traditional methods of diagnosing depression include consulting with an expert. However, diagnosis of psychological depression involves human subjectivity. In addition, people are reluctant to reveal their condition to experts due to social stigma. Therefore, there is a need for a model to diagnose depression in an objective and consistent way. Models use brain waves and machine learning methods; hence, they have gained attention as they are objective and do not depend on experts. This study proposes a deep-asymmetry method, which uses an EEG-based image asymmetry matrix along with a CNN. The proposed methodology and classification model classified MDD patients and healthy individuals with an accuracy of 98.85%, which exceeds accuracy of most existing methods. The proposed method extracts features by preserving the spatial characteristics of the EEG channel. In addition, the proposed process is performed automatically without a manual feature extraction method. Such an automatic method can be used by nonprofessionals and could be used as an effective tool for pre-screening depression.

## Figures and Tables

**Figure 1 sensors-20-06526-f001:**
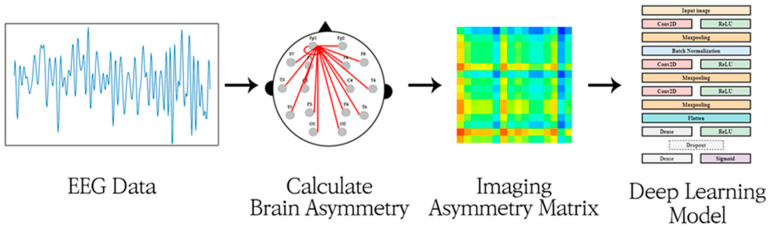
Illustration of the deep-asymmetry method.

**Figure 2 sensors-20-06526-f002:**
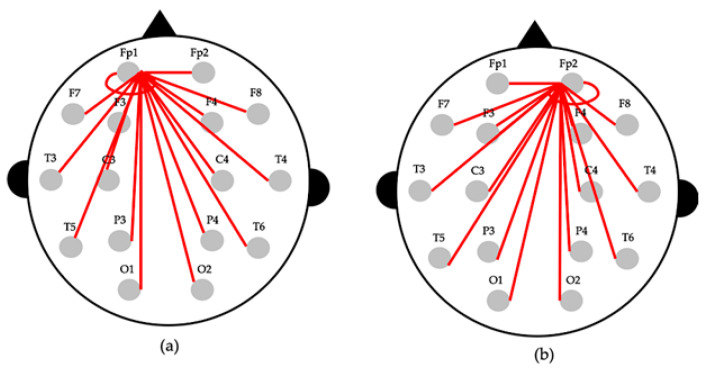
Example for the calculation of an EEG Asymmetry. (**a**) asymmetry between Fp1 and the other channels. (**b**) asymmetry between Fp2 and the other channels.

**Figure 3 sensors-20-06526-f003:**
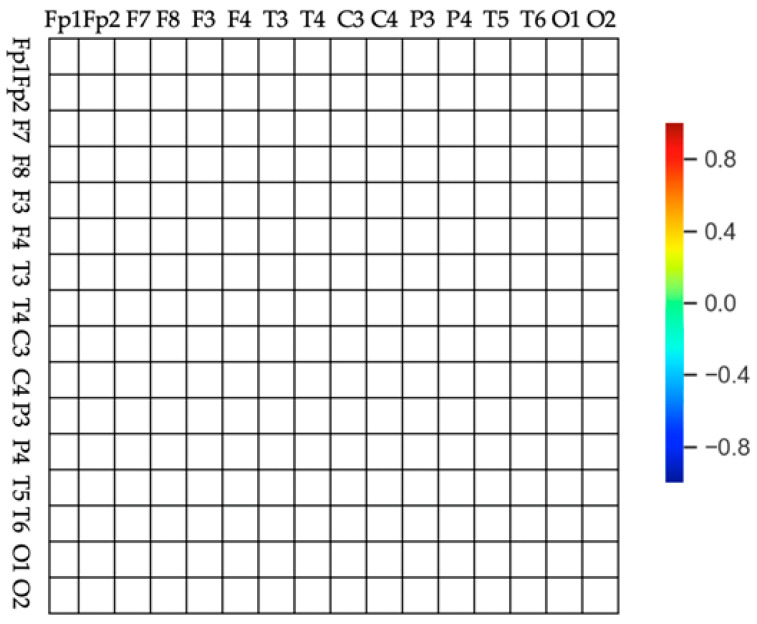
Frame of the brain asymmetry image. The rows and columns correspond to the 16 channels (Fp1, F3, C3, P3, O1, F7, T3, T5, Fp2, F4, C4, P4, O2, F8, T4, and T6) and every cell has a color based on the value of A (ch1, ch2), as indicated on the color bar on the right side.

**Figure 4 sensors-20-06526-f004:**
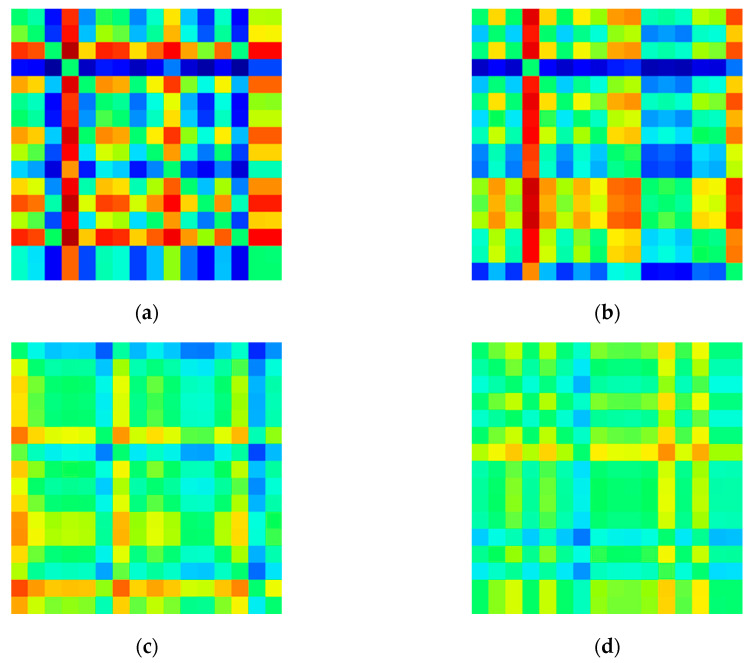
Visualized EEG images for healthy controls at various frequency bands in EC dataset: (**a**) delta (**b**) theta (**c**) alpha, and (**d**) beta.

**Figure 5 sensors-20-06526-f005:**
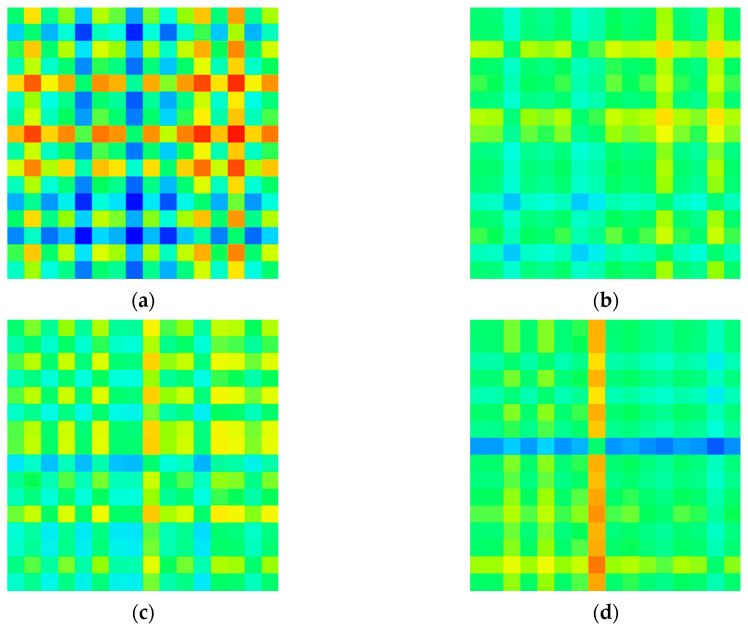
Visualized EEG images for MDD patients at various frequency bands in EC dataset: (**a**) delta, (**b**) theta, (**c**) alpha, and (**d**) beta.

**Figure 6 sensors-20-06526-f006:**
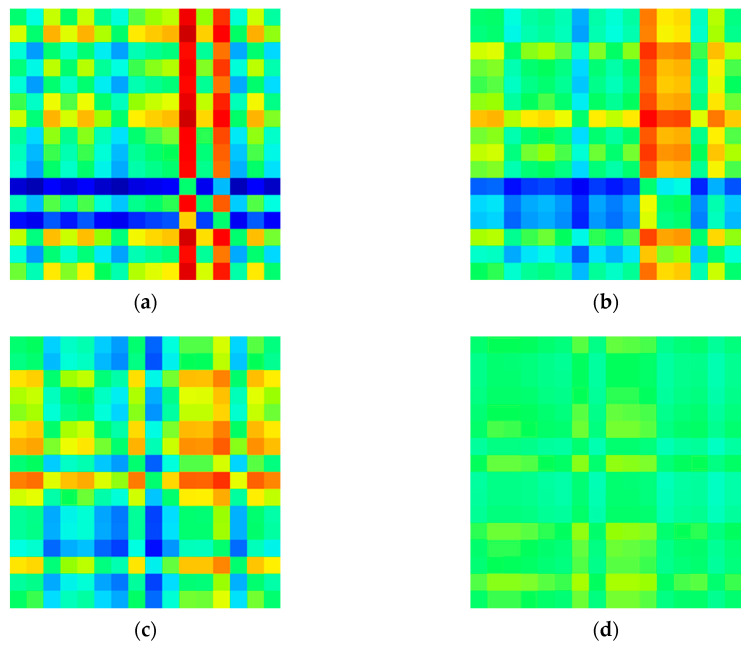
Visualized EEG images for healthy controls at various frequency bands in eyes-opened(EO) dataset: (**a**) delta (**b**) theta (**c**) alpha, and (**d**) beta.

**Figure 7 sensors-20-06526-f007:**
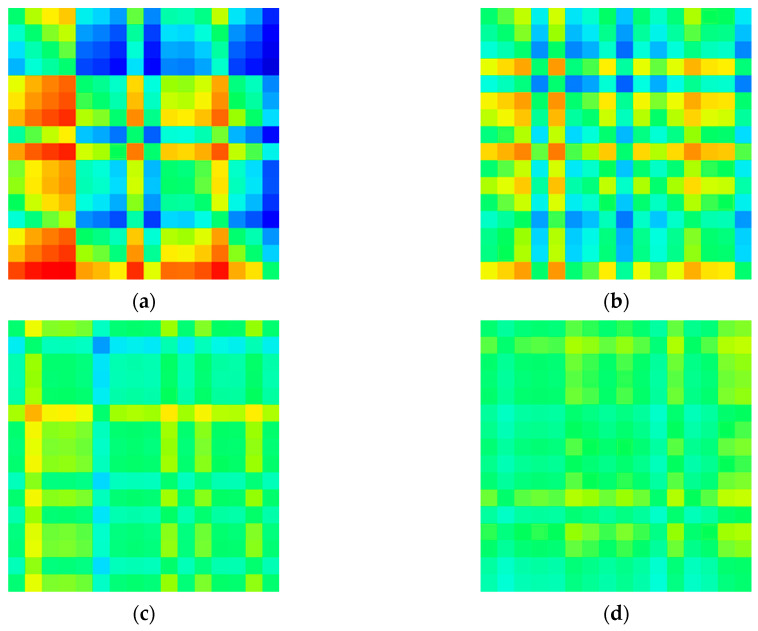
Visualized EEG images for Major Depressive Disorder (MDD) patients at various frequency bands in EO dataset: (**a**) delta, (**b**) theta, (**c**) alpha, and (**d**) beta.

**Figure 8 sensors-20-06526-f008:**
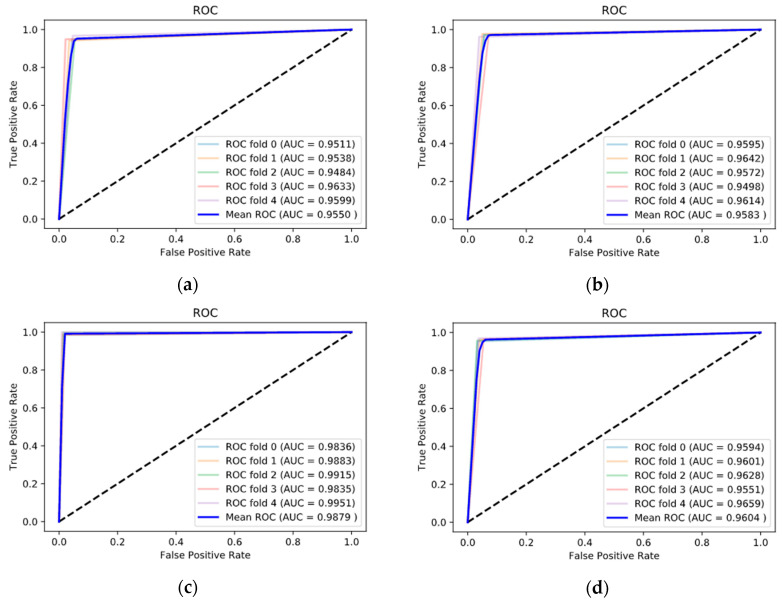
Receiver operating characteristic (ROC) curves of the deep learning model in EC dataset: (**a**) delta, (**b**) theta, (**c**) alpha, and (**d**) beta.

**Figure 9 sensors-20-06526-f009:**
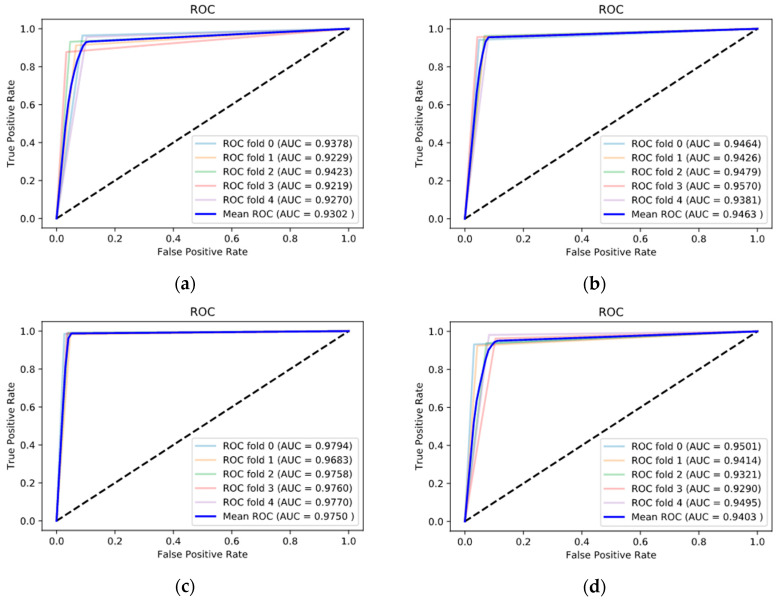
ROC curves of the deep learning model in EO dataset: (**a**) delta, (**b**) theta, (**c**) alpha, and (**d**) beta.

**Table 1 sensors-20-06526-t001:** Parameters of the CNN model.

Layer	Output Shape	Param #
Input	64, 64, 3	-
2D Convolution (C1)	62, 62, 32	896
2D Maxpooling (P1)	31, 31, 32	0
Batch normalization (BN)	31, 31, 32	128
2D Convolution (C2)	29, 29, 64	18,496
2D Maxpooling (P3)	14, 14, 64	0
2D Convolution (C3)	12, 12, 128	73,856
2D Maxpooling (P3)	6, 6, 128	0
Flatten	4608	0
Dense 1	256	1,179,904
Dropout (DP)	256	0
Dense (sigmoid)	1	257

**Table 2 sensors-20-06526-t002:** Results of proposed model in eye-closed (EC) dataset.

Band	Accuracy	Sensitivity	Specificity
Delta	95.50	95.11	95.95
Theta	95.90	97.11	94.56
Alpha	**98.85**	99.15	98.51
Beta	96.07	96.13	96.00

**Table 3 sensors-20-06526-t003:** Results of proposed model in EO dataset.

Band	Accuracy	Sensitivity	Specificity
Delta	93.03	92.83	93.24
Theta	94.66	95.41	93.86
Alpha	**97.56**	98.75	96.31
Beta	94.06	94.75	93.33

**Table 4 sensors-20-06526-t004:** Overview of related studies.

Studies	Methods	Classification Methods	Accuracy
Wajid Mumtaz (2016) [12]	Frequency power + asymmetry feature	SVM	98.4
Shalini Mahato (2018) [14]	Alpha power + RWE	MLPNN	93.33
Wajid Mumtaz (2019) [18]	raw EEG	1D CNN	98.32
Shalini Mahato (2019) [13]	Alpha power + theta asymmetry	SVM	88.33
Abdolkarim Saeedi (2020) [35]	Effective Connectivity (GPDC, dDTF)	1D CNN + LSTM	99.25
Current study	Asymmetry image	2D CNN	98.85

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
