# Peer review of "Deep-Asymmetry: Asymmetry Matrix Image for Deep Learning Method in Pre-Screening Depression"

_sensors, 2020, doi:10.3390/s20226526_

Round 1

Reviewer 1 Report

Actually, this research is not that far from I am personally working on so restI can clearly state that I am quite aware of signifficance of this publication.

What the Authors have done is that they developed a method to run pre-screening tests and use deep learning methods to detect depression. This kind of research is of really high importance because having a smart system that would at least identify some ambiguous test results indicating problems on which doctors could have a look rather that go through all tests. By proportion, there is less people with various kinds of mental problems than entirely healthy people so as far as doctor's time usage and expertise are concerned, going through all tests results rather than focusing on potentially problematic ones is of utmost importance.

This paper provides a sound research and quite clearly described methodology. What I would like to see in a bit more detail (but only to better understand the diagnostic process from the beginning to the end) about the ANN structure / specifics. How it was taught, how was its operation verified, especially for some newly acquired, on-line, data rather than data coming from a database. But other than that I have no reservation over the paper quality and would recommend publishing it as it is (use of English is really good and hence the paper does not require any signifficant changes in this regard). 

Reviewer 2 Report

Comments

This is a small study. There is a risk of overfitting. It is interesting because it converts the EEG features into a matrix, which should make it more understandable to readers. However, the authors do not comment on the visually obvious differences in Figure 4(c) and Figure 5(c) which form the core of their study. 

I have the following concerns:

  • the authors used only one small dataset which has already been used in other papers.
  • the data appear to be driven by less asymmetry in posterior channels in depressed patients. Does this make biological sense?
  • The authors used the eyes closed dataset only, which would give them more posterior dominant rhytm. Figure 4(c) and Figure 5(c) indicate differences likely due to the posterior dominant rhytm. The authors should also evaluate the eyes open portion. 
  • There would be less posterior dominant rhytm in the eyes open dataset, which would likely give less diagnostic performance, and the paper would be more useful/believable/explainable with this.
  • Using only a minimal portion of a small dataset gives the impression of low effort, though this may not be the case.
  • The EEG data processing pipeline is incompletely described. Was EEGLAB used? The paper is unusually unhelpful to other researchers wanting to replicate their study. If the authors wanted to be really pro-active, they could publish their code on GitHub. This would be a big step towards showing that the authors themselves trust their pipeline.
  • There is a crisis of reproducibility in neurophysiology. The authors should be much more clear that this is a small pilot study, acknowledging at least the risk of overfitting and the need for replication. 
  • The authors might do well to study the correlation of their proposed classification feature (the final output of their neural network, some sort of linear feature which the ROC curves are based on) with clinical features and scores in the dataset such as BDI and HADS. Any correlation would make the paper more believable.
  • I encourage the authors to get advice from a clinical  EEGer and/or a psychiatrist. 

In sum, the paper needs more work to explain how they processed the data, and what underlying feature of depression their proposed EEG feature explains.

Reviewer 3 Report

The authors present an interesting new method for converting spectral features of an EEG to Correlation Matrix of Asymmetry. This kind of images are suited for machine learning algorithms using deep learners. Data was collected from a previously published open database. Methods for EEG processing seem to be robust, and statistical results are sound. ROC AUC are really strong. My comments are: I would recommend revising the introduction, the paper is well written and complex, but the introduction is a bit simplistic, the authors should stress the usefulness of having a quantitative neurophysiologist tool for depression, since diagnosis is so subjective. Moreover, MMSE Minimental is not an instrument for diagnosis in Depressive diseases, in that setting it is used to make differential diagnosis with cognitive impairment. Welch's periodogram-> Please specify window length and overlap. Please specify sampling rate used for recordings, i suppose it was 256 Hz Is it feasible to evaluated if the use of alpha and delta maps combined yield greater classification power. Describe how diagnosis of depression was made in the original database. Since alpha asymmetry is such a powerful tool to evaluate depression in would be interesting to plot BDI scores compared with average Alpha asymmetry among channels, or with some other form of data transform that explains the degree of asymmetry. I think that another issue that should be addressed is that form of deep learning make it hard for us to understand which factors contribute the most to the classificator, we can suppose that the asymmetry is the greatest contributor, but we cannot exclude that asymmetry in some positions have great weight in the decision.

Reviewer 4 Report

This article presents a new method to detect depression. It combines an EEG asymmetry with CNN. In general, this method is well explained. However, I have some comments and questions:

  1. Line 94. The sentence " For removing blink noise simply, the EC data was used". I think this phrase could be removed. The meaning and the CE symbol are not understood.

  2. Section 2.2: You say that ICA is used to remove artifacts. Could you explain the number of independent components that are used to reconstruct the EEG channels without artifacts?

    In general, after using ICA, the reconstructed EEG signal loses some information in the artifact-free signal segments. Can this loss of information have a positive impact on your algorithm?

  3. In Fig. 3:  You have decided to build an asymmetry image using a color image. Why did you decide to use a color image instead of a gray image? In this case, I think a gray image would have the same amount of information as its color image.

  4. Regarding depression, is it possible to characterize the severity of this disease using a scale? In this case, is your method helpful in detecting mild depression?

  5. Section 2.5. Could you explain the size of the subset for cross validation? Although cross-validation is useful, evaluation data is typically performed using training and test data sets. Don't you have enough data for this evaluation?

  6. Line 208:"Similar to other studies, the size of the data set is the limitation of the study. To overcome this problem, data segmentation, batch normalization, and dropout layer were used in this study. Future research will be focused on generalizing results and presenting a more reliable model using more samples and other types of EEG data. ". Why do you think the size of your data set is small? Do you think your data set does not cover all levels of depression? Are you worried that your model will generalize badly?

Round 2

Reviewer 2 Report

The disclosure of the code is reassuring. I still have concerns about overfitting.

An appropriate control experiment can be derived from the freely available Temple University Hospital EEG corpus at https://www.isip.piconepress.com/projects/tuh_eeg/html/overview.shtml

The authors could show that their experiment is not due to overfitting. They could download 64 EEGs described as normal from this corpus. They could randomly assign a "depressed" status to 34 of these, and 30 to normal status. They could then run this control set through their pipeline. The AUC should be close to 0.5, unless overfitting is present. This experiment could be done five times. The average AUC should be close to 0.5.

The authors could also evaluate a number of these control EEGs using their already developed algorithm, and estimate what percentage of these EEGs are classified as depressed. The prevalence of depression in this corpus is unknown, but it would be unexpected that it would be very high. If the prevalence in this control set is close to 50%, like in the development set, then the result is also likely to be due to overfitting. 

Author Response

Thank You.

Reviewer 4 Report

OK about your answers. I encourage the authors to continue this research. it is worthwhile.

Author Response

Thank you very much for your appreciation of our research.

We have worked hard to incorporate your feedback and hope that these revisions persuade you to accept our submission.

Sincerely yours.

Corresponding Author’s Information

- Name : Young-ho Lee

- Affiliation : Dept. of Computer Engineering, Gachon University, Korea

- E-Mail : lyh@gachon.ac.kr

- Phone : +82.10.5005.2275